# Assessing the Sustainable Room for Growth for a Particular Romanian Tourism Area of Business: The Case of Accommodation Businesses

**Sorin-Romulus Berinde [1] and Adrian-Gabriel Corpădean [2,\*]** 

1    Faculty of Business, Babeş-Bolyai University, Cluj-Napoca, Romania, 7 Horea Street, Cluj-Napoca, Cluj 400174, Romania; sorin.berinde@tbs.ubbcluj.ro

2    Faculty of European Studies, Babeş-Bolyai University, Cluj-Napoca, Romania, 1 Emmanuel de Martonne Street, Cluj-Napoca, Cluj 400090, Romania

\*    Correspondence: adi_corpadean@yahoo.com; Tel.: +40-741-249-141

**Abstract:** In recent years, the Romanian accommodation business sector has recorded a dramatic increase in accommodation units, but even under these circumstances it is below half of the average registered by the European Union. The study aims to evaluate whether there is still room for sustainable growth in this regard. For this purpose, the accounting–financial indicators have been assessed for an annual average of 3447 companies reporting every year to the European Union for 18 years, between 1999 and 2016, for all the accommodation units in Romania and the European Union. For data processing and to assess sustainable growth, we have used simple regression and resorted to the distance method and the geometric mean method to analyze competitiveness. The findings show a likelihood of sustainable growth of 20.6% in the development of Romanian accommodation businesses, in the light of the aspects analyzed, correlated to the EU average. Some managerial decision-making suggestions are provided at the end of the paper for accommodation businesses' sustainable growth, related to accounting–financial issues. For Romanian businesses, sustainable growth is promoted by the low level of staff costs and, to a lesser extent, by the investments made per employed person. For the corporate governance of these companies to recover growth space in terms of sustainability, managerial decisions should be taken to increase sales, profitability, production value and added value.

**Keywords:** accounting–financial issues; sustainable growth; accommodation businesses

## 1. Introduction

The tourism area in general and the accommodation sector in particular have been listed as showing an accentuated upward trend in Romania in recent years. Although between 1999 and 2016, the average number of companies which offered accommodation services in Europe slowly increased by an average of 35%, in Romania, the number of accommodation units increased by 7.31 times. Even under these circumstances, Romania had only 47.03% of the average number of accommodation units existing at the European Union (EU) level in 2016, as shown in Table 1.

This exponential increase in the number of accommodation units in Romania should be correlated with the fact that it did not even attain half the average number of accommodation units in the EU, according to the Eurostat data evolution presented in Figure 1a. At the same time, as shown in Figure 1b, the average number of employees in the Romanian accommodation sector declined much faster than the European average, reaching very near to the levels in 2016. In these circumstances, it becomes of interest to conduct research aimed at studying the extent to which there is room for sustainable growth in the case of Romanian accommodation businesses. Given that the increase in the

number of Romanian accommodation businesses is so significant, it is very interesting to assess how much these companies could grow.

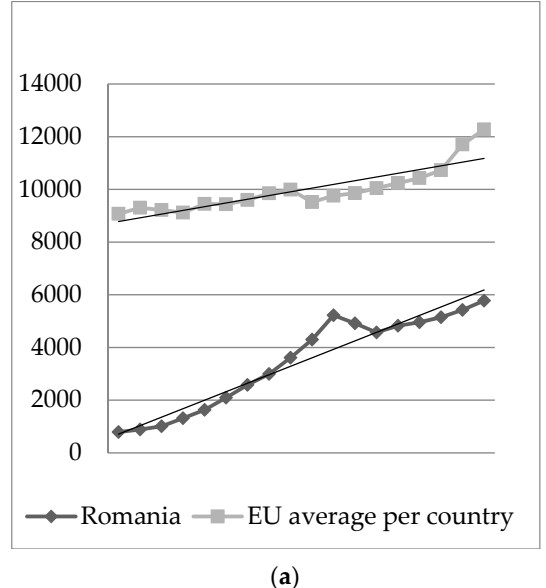 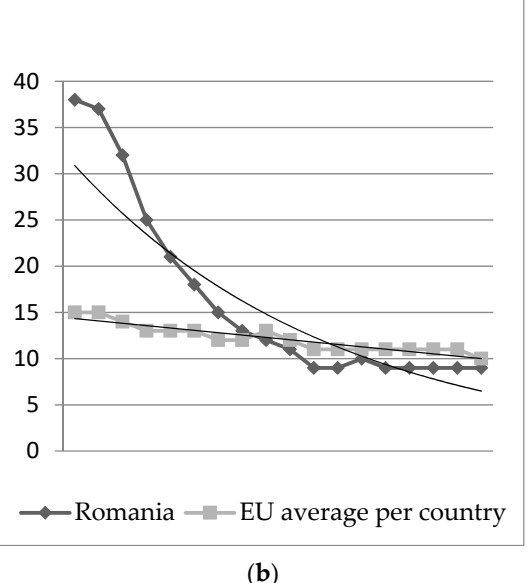

(**a**)                                                (**b**)

**Figure 1.** (**a**) Evolution of accommodation business units between 1999 and 2016; (**b**) evolution of the average number of employees per business between 1999 and 2016.

**Table 1.** The evolution of the number of companies and the average number of employees per company in the accommodation sector in Romania related to the EU average.

| | 1999 | 2000 | 2001 | 2002 | 2003 | 2004 | 2005 | 2006 | 2007 | 2008 | 2009 | 2010 | 2011 | 2012 | 2013 | 2014 | 2015 | 2016 |
|---|---|---|---|---|---|---|---|---|---|---|---|---|---|---|---|---|---|---|
| **Evolution of the number of companies** | | | | | | | | | | | | | | | | | | |
| Romania | 790 | 891 | 1010 | 1316 | 1632 | 2094 | 2578 | 2997 | 3611 | 4297 | 5222 | 4918 | 4573 | 4824 | 4955 | 5149 | 5422 | 5774 |
| EU average | 9070 | 9301 | 9214 | 9116 | 9449 | 9438 | 9597 | 9854 | 9988 | 9517 | 9757 | 9855 | 10,045 | 10,236 | 10,430 | 10,728 | 11,705 | 12,277 |
| **Evolution of the average number of employees per company** | | | | | | | | | | | | | | | | | | |
| Romania | 38 | 37 | 32 | 25 | 21 | 18 | 15 | 13 | 12 | 11 | 9 | 9 | 10 | 9 | 9 | 9 | 9 | 9 |
| EU average | 15 | 15 | 14 | 13 | 13 | 13 | 12 | 12 | 13 | 12 | 11 | 11 | 11 | 11 | 11 | 11 | 11 | 10 |

Source: Data provided by Eurostat (Directorate-General of the European Commission) between 1999 and 2016.

The objective of this research is to measure and to present the perspectives for the sustainable growth of Romanian accommodation businesses based on an 18-year period (1999–2016) in a single general ratio which incorporates a mix of relevant issues that have been assessed. This ratio will show how much Romanian accommodation businesses could grow sustainably when the average level recorded for accommodation businesses at the European Union level is considered to be the ideal standard.

This study applies the most appropriate simple regression methods for measuring sustainable growth; i.e., the ones based on a comparative analysis of evolution in time. The study extends previous macroeconomic research on sustainable growth to microeconomic aspects, particularly related to the Romanian accommodation businesses from the tourism area. The variables for measuring growth perspectives are quantitative, closely related to financial–accounting aspects in general, and particularly to profit, sales, employment, economic added value and investment. These issues are considered appropriate for sustainable growth measurement according to previously published research.

Relevant literature includes studies that test, by various methods, the correlation between a certain category of parameters which determine sustainable growth and another category of parameters that measure sustainable growth itself [1]. These studies are generally accompanied by the confirmation or

refutation of certain hypotheses related to the impact that these two categories of parameters have on growth: positive, negative or no impact (neutral).

There is a gap in the literature in terms of assessing sustainable growth for a particular tourism area; i.e., accommodation businesses. Additionally, this is done in a comparative way, as the difference between the levels of growth registered in time is measured by comparing two particular categories of companies. This study aims to measure the distance between the parameters that define growth for two categories of companies in the accommodation area: Romanian companies and the EU average in this respect. The distance method has been used to assess the differences in relative sizes. Subsequently, a method for the aggregation of the calculated distances has been proposed in order to establish a representative quantitative value for assessing the difference between the two categories of company growth, and also for assessing the sustainable room for growth for the second category of companies, which have recorded lower growth.

This paper is organized as follows: the present section gives an introduction. Section 2 exhibits a relevant review of literature. Section 3 displays the research methodology supporting the study. Section 4 reports the results. Section 5 presents the discussions. Finally, Section 6 concludes the paper.

## 2. Literature Review

Studies in the literature addressing the influence of different issues on growth are carried out mostly at a macroeconomic level [2–4]. On the other hand, highlighting economic growth as an impact of financial development can be better emphasized by using time-series than by conducting cross-country studies [5]. In this macroeconomic context, the accounting–financial impact on sustainable growth is achieved through four channels: production—which is the most important— savings, fixed investments and schooling [6]. This study is based on data collected for 21 OECD countries over the past 140 years. The impact of investments on growth at the macroeconomic level is an aspect that has often been studied in the literature [7–10]. Macroeconomic growth can also be sustained in the long term by profit maximization, the increasing returns in the production of outputs and the decreasing returns in the production of knowledge [11]. The impact of labor productivity on growth has been the subject of recent studies [12,13].

The study of sustainable growth at a microeconomic level can be achieved under various aspects: individual, firm and industry. Furthermore, there are other methods of estimating a company's growth apart from measuring its turnover [14]. Irrespective of the aspect under which growth appears at the microeconomic level, one can consider that profit sizing has a vital role in a company's growth [15].

The most frequent studies approach the sustainable growth of companies that are part of a particular industry or are included in the same size category. The field of activity of the companies included in the study could have a significant impact on the results when analyzing the companies' growth due to certain issues. In this respect, there are studies which refer especially to the service sector [16,17], to the manufacturing sector [18], or to the non-manufacturing sector [1,19,20], to online trade [21], or to the field of companies listed on the stock market [22]. There are relatively few studies which have been conducted on sustainability in tourism and which, at the same time, attempt to evaluate sustainable growth using quantitative accounting–financial indicators [23–26].

At the microeconomic level, companies' sustainable growth has been measured from a quantitative perspective by certain authors by using data from accounting–financial statements. Growth may be supported by an absolute or a relative variation of turnover, or the number of employees, age, number of companies, size and industry [27–30]. The assessment of sustainable growth exclusively in terms of turnover could have a diminished degree of relevance due to the inflationary impact over time. This effect is less noticeable in countries with low inflation. The higher the time range included in the study is, the greater this risk of distortion caused by inflation becomes; but on the other hand, there is also an advantage linked to the increasing relevance of the information obtained from processing a larger sample.

Other studies consider that sustainable growth can be measured by companies' ability to accumulate resources, including labor [31,32]. The shareholding structure is another element considered for measuring growth capacity [33]. An increase in the independence of the companies that are not controlled by the parent company generates greater flexibility in identifying opportunities. On the other hand, these companies, since they are relatively small, may have difficulties in financing investment opportunities. The capacity of governance to make decisions, motivate staff and establish objectives, the level of education and the managers' experience are aspects that can generate growth in companies [34–36]. These managerial capabilities are themselves influenced by the dynamics of the market on which they operate and the availability of capital for investment [37,38].

In small companies in particular, the growth potential is significantly conditioned by access to external funds, especially indebtedness by contracting bank loans [39,40]. For small companies, it is otherwise difficult to turn to domestic financing sources, which are cheaper, and secured by the capital market. Even if the costs of external financing are higher, small companies manage to grow more than large ones, by using indebtedness from external sources.

Some research results show that women-owned businesses record lower growth or have lower growth aspirations [41]. Setting up new activities or ones similar to those being currently carried out, and creating new products and services, can be considered aspects of entrepreneurship that generate growth [42,43].

Financial performance, measured by the level of profit and profitability, is linked to sustainable growth [44–46]. Companies' sustainable growth is correlated to the same extent whether we refer to the profitability from the previous period or to the level of profitability from the current period [47]. Even though making profit is an aspect that undeniably supports companies' growth, it is not sufficient to ensure the latter [48].

The impact of company size on sustainable growth has been studied in the literature from various perspectives. If, when we refer to company size, we take sales, the number of employees, fixed costs, profit and liquidity into consideration, it seems that small companies grow faster than large ones, given that the indicators analyzed register positive evolutions [47,49,50]. In addition, conflicting results regarding the impact of the number of employees and profit on growth were registered only in 10% of limited liability companies in Sweden [51–54].

If diversification occurs within narrow limits, in related fields, it can also lead to sustainable growth [55,56]. For new companies, activity diversification is detrimental to efficiency, but also provides prerequisites for growth and survival in this critical phase of their existence [57]. Networking can refocus companies from descending trends to growth and productivity [58].

Innovation in itself, although not sufficient to induce substantial growth [54], can support the growth of larger companies with experience, while this option is riskier for small companies [59]. Setting aside the age of companies, it is clear that companies that innovate grow faster than those that do not [60–62]. On the other hand, the impact of innovation on growth is higher if the costs are financed from domestic sources, as opposed to seeking funding from external sources [63].

Companies that rely on their own financial resources grow at a slower pace because it is more difficult for them to make investments. Resorting to indebtedness and incurring additional financing costs ensure more feasible funding, which induces sustainable growth in companies [64]. On the other hand, implementing an adequate control system of added value leads to a more efficient use of resources, an increase in investment capacity and, thus, to a better situation for companies with slow growth [65]. For venture capital-backed firms in high-tech European industries, growth is mostly supported by the added value created by the companies rather than by screening [66]. Besides this, the impact of added value on growth at the microeconomic level has been studied very little in the literature.

The impact of the share capital structure on a company's sustainable growth is studied from the perspective of family-owned businesses, of the dispersion of ownership and foreign ownership, and presents mixed outputs.

Family-owned businesses, with a low diversification of share capital, exhibit some reluctance to hire staff, which significantly reduces their growth capacity [67]. There are also results which show that most family businesses simply do not grow [68]. To support the sustainable growth of family-owned businesses, one must consider making changes in the business life cycle, the organization and the chief executive [69]. Other studies argue that there is no link between growth and family or non-family business-owned companies [54].

Dispersed shareholding exhibits greater growth potential, as corporate governance has a more complex contribution in this respect [70]. On the other hand, having several shareholders means benefitting from more ideas and therefore more disagreements with regard to business strategy, which can lead to a negative growth effect [54].

The presence of foreign shareholders in the structure of share capital supports the hypothesis of the company's sustainable growth [71]. This hypothesis is explored in other studies. Foreign owners have a negative impact on employment increases, a positive effect on equity increases, and, overall, the inclusion of foreign shareholders in the ownership structure would have a negative impact on composite growth rate [54].

Firm sustainable growth patterns are briefly shown below in Table 2.

**Table 2.** The aspects considered in the literature for assessing sustainable growth.

| Firm Growth Patterns | Author(s) of the Study and Year of Publication |
|---|---|
| Industry | Megaravalli, 2017; Gao et al., 2016; Gao et al., 2015; Tang, 2015; Adams et al., 2014; Bhattacharyya and Saxena, 2009; Lee, 2009; Audretsch et al., 2004 |
| Sales growth, employee growth | Babalola, 2013; Delmar et al., 2003 |
| Age, legal form, number | Fiala and Hedija, 2015; Bornhall et al., 2013; Bentzen et al., 2012; Delmar et al., 2003; Caroll and Hannan, 2001 |
| Resource and knowledge view | Penrose 1995; Kogut and Zander 1992 |
| Features of management style (managerial skills, attitude towards growth, gender effect) | Szerb and Ulbert, 2006; Baum and Locke, 2004; Wiklund and Shepherd, 2003; Du Rietz and Henrekson, 2000; Barney, 1991 |
| Entrepreneurship | Gartner and Carter, 2003; Davidsson et al., 2006 |
| Market conditions, the availability of capital | Wiklund and Shepherd, 2005; Carlsson, 2002 |
| Indebtedness level (equity, liabilities) | Pervan and Višić, 2012; Becchetti and Trovato, 2002; Winborg and Lanstrom, 2001 |
| Internationalization | Julien and Ramangalahy, 2003 |
| Financial performance (profits, profitability) | Perényi and Yukhanaev, 2016; Demirgunes and Ucler, 2015; Davidsson et al., 2013; Mukhopadhyay and Amir Khalkhali, 2010; Cowling, 2004 |
| Firm size (annual sales, number of employees, asset value, productivity, fixed costs, liquidity) | Iribarren et al., 2016; Perényi and Yukhanaev, 2016; Fiala and Hedija, 2015, Bornhall et al., 2013; Daunfeldt and Elert, 2013; Bentzen et al., 2012; Reid, 2007; Szerb and Ulbert, 2006 |
| Investments | Wang and Wang, 2016; Szerb and Ulbert, 2006 |
| Diversification, networking | de Andrés et al., 2017; Mendoza-Abarca and Gras, 2017; Sahu, 2017; Benson et al., 2016 |
| Innovation, R&D | Coad et al., 2016; Bianchini et al., 2016; Audretsch et al., 2014; Segarra and Teruel, 2014; Inzelt and Szerb, 2006 |
| Economic value added | Wang and Wang, 2016; Croce et al., 2013 |
| Social capital (family businesses, dispersion of ownership, foreign investments) | Storey, 2016; Ward, 2016; Szerb and Ulbert, 2006; Cromie et al., 1999; Ward, 1997 |

Source: Processing performed by authors using literature research studies.

These firm growth patterns related to growth shown above and used in the literature are split into two categories: the first includes issues that have an influence on business growth and the second comprises issues that measure growth [56], which are ranked and shown below in Table 3.

**Table 3.** A synthesis of growth patterns in the literature. Description and classification.

| | | |
|---|---|---|
| **Patterns with an impact on business growth** | **Personal issues** | Family businesses, dispersion of ownership<br>Features of management style<br>Entrepreneurship |
| | **Business issues** | Industry<br>Age, legal form, number<br>Internationalization<br>Resource and knowledge view<br>Market conditions, availability of capital<br>Diversification, networking<br>Innovation, R&D<br>Foreign investments |
| **Patterns for assessing business growth** | | Sales growth<br>Employee growth<br>Indebtedness level (equity, liabilities)<br>Financial performance (profits, profitability)<br>Asset value, productivity, fixed costs, liquidity<br>Investments<br>Economic value added |

Source: Processing performed based on Szerb and Ulbert's conceptual model related to firm growth from 2006 [56].

The current research is conducted under the auspices of quantitative patterns for assessing businesses' sustainable growth. Taking this into account, the study will turn to appropriate items used in the literature for measuring business growth in order to assess the room for the sustainable growth of Romanian accommodation businesses.

Opinions are divided in the literature regarding the variety of methods used to measure growth. Some authors promote the idea of resorting to a single measurement method [72–74]. They argue in favor of this by stating that the aggregation of several different measurement methods leads to the irrelevance of the study, because each method affects growth in different ways and to varying degrees. On the other hand, applying several methods of measuring companies' growth is likely to reduce estimation errors, increase robustness and ensure a greater comparability of the results obtained from processing the data with different methods [29]. However, comparability can be ensured only if the measurement methods used are compatible. We believe that comparability is justified to a great extent by a long-enough time interval and a sample that is large enough to ensure the consistency and relevance of the results. In this context, to increase relevance, we consider that it is more appropriate to use a variety of indicators and methods for measuring a company's sustainable growth, rather than processing it through a small number of indicators or by a single method. To remove the influences of different accounting and taxation systems at the national level for each European country assessed in this comparative analysis of evolution in time, one can use methods that provide outputs in relative sizes.

## 3. Materials and Methods

The variables taken into consideration for measuring sustainable growth are of a quantitative nature, obtained from the financial reports published and reported to the EU by an annual average of 3447 companies operating in the Romanian accommodation area, for each year between 1999 and 2016. The quantitative data is more accurate, measurable, reliable, and enables a more precise quantification of sustainable growth: sales, employees, liabilities, profitability, productivity, investments and economic value added. In order to measure sustainable growth, a number of relevant quantitative

variables have been associated with each growth pattern, as shown in Table 4 below, while taking into account the mixture of variables used in the literature to measure each pattern of business growth. Considering the fact that the selected variables are more developed and more complex than in any previously conducted study, we are entitled to say that the growth measured is sustainable.

**Table 4.** The connection between patterns and variables.

| Patterns for Assessing Business Growth | Quantitative Variables Proposed for Assessing Sustainable Growth |
|---|---|
| Sales growth | Turnover per person employed |
| | Number of unpaid persons employed |
| Employee growth | Number of employees |
| | Average personnel costs |
| | Growth rate of employment |
| Indebtedness level (equity, liabilities) | Total purchases of goods and services |
| Financial performance (profits, profitability) | Gross operating surplus |
| | Gross operating rate |
| | Share of gross operating surplus in value added |
| Asset value, productivity, fixed costs, liquidity | Production value |
| | Apparent labor productivity |
| | Share of personnel costs in production |
| | Share of personnel costs in total purchases of goods and services |
| Investments | Investment per person employed |
| | Investment rate |
| Economic value added | Value added at factor cost |
| | Gross value added per employee |
| | Value added at factor cost in production value |

Source: Processing performed by the authors.

The study relies on the data provided by Eurostat [75] for Romanian and EU accommodation businesses: hotels, camping sites and other forms of provision of short-stay accommodation. Data is collected separately, for a time span of 18 years, between 1999 and 2016, for Romania (dependent variable) and the average per country recorded at EU level by leaving out Romanian data (independent variable). For each year of the aforementioned period, an annual average number of 3447 Romanian companies and an annual average of 9977 EU companies per country were included in the comparative study. The variables to be taken into consideration are quantitative, of a financial and accounting nature, as shown in Table 5. The Romanian level was considered as a dependent, and the EU level as an independent variable for each year during the 18 years when the study was conducted. The variables are closely related to patterns for assessing/measuring business growth, as presented in Table 4 above.

The study is conducted based on the assumption that from the diversity of variables presented, only those for which an acceptable level of correlation will be established between the temporal evolution of tourism companies in Romania (dependent variable) and the average recorded at the EU level (independent variable) will be taken into consideration for measuring the room for sustainable growth. The linear regression model shown in Equation (1) was used:

$$Y = a + b \, x \, X \tag{1}$$

where $Y$ is the dependent variable, $X$ is the independent variable, and $a$, $b$ are the regression model parameters.

In order to measure the distance for Romanian accommodation businesses compared to the EU average level, the distance method was used [76]. This process involves taking the recorded values of each variable into account for each year within the 1999–2016 range. In this way, it assesses in relative values the distance measurement between the levels achieved by the Romanian companies from the accommodation area, compared to those registered in the EU, as an average per country for the whole period between 1999 and 2016. To avoid cumulating similar values and distorting the

results, the formula was completed to obtain an average value of all annual distances measured in the analyzed time span. The distance measurement is carried out as follows and shown in Equation (2):

$$D(Vj) = \sqrt{\frac{\sum_{i=1}^{n}\left(100 - \frac{Vji(k)}{Vji(e)} x100\right)^2}{n}} \tag{2}$$

where $D(Vj)$ is the distance for each selected variable $j$ between the Romanian and EU average level, where the values selected for $j$ can be $j = \overline{1, \ m}$, and $m$ represents the number of selected variables, with a single one for each of the seven patterns; $Vji(k)$ is the value of the selected variable $j$, from year $i$, for the Romanian companies $k$, and $i = \overline{1, \ n}$, where $n$ represents the number of years for which the study is conducted; and $Vji(e)$ is the value of the selected variable $j$, from the year $i$, for the average recorded for companies $e$ in the EU, $i = \overline{1, \ n}$, where $n$ represents the number of years for which the study is conducted.

**Table 5.** Description of variables.

|    | Variable Explanation | Variable Symbol | Measuring Unit |
|----|----------------------|-----------------|----------------|
| 1  | Turnover per person employed | TurnPersEmpl | Thous. Euro |
| 2  | Number of unpaid persons employed | NoUnpPersEmpl | No. of persons |
| 3  | Number of employees | NoEmpl | No. of persons |
| 4  | Average personnel costs | AvPersCosts | Thous. Euro |
| 5  | Growth rate of employment | GrowRateEmpl | % |
| 6  | Total purchases of goods and services | TotPurchGS | Mil. Euro |
| 7  | Gross operating surplus | GrossOperSurpl | Mil. Euro |
| 8  | Gross operating rate | GrossOpRate | % |
| 9  | Share of gross operating surplus in value added | ShareGrossOpSurplValAdd | % |
| 10 | Production value | ProdVal | Mil. Euro |
| 11 | Apparent labor productivity | AppLabProduct | Thous. Euro |
| 12 | Share of personnel costs in production | SharePersCostProd | % |
| 13 | Share of personnel costs in total purchases of goods and services | SharePersCostPurch | % |
| 14 | Investment per person employed | InvestPersEmpl | % |
| 15 | Investment rate | InvestRate | % |
| 17 | Gross value added per employee | GrossValAddEmpl | Thous. Euro |
| 16 | Value added at factor cost | ValAddFactCost | Mil. Euro |
| 18 | Value added at factor cost in production value | ValAddProdVal | Mil. Euro |

Source: Processing performed by the authors using data provided by Eurostat.

Since, after establishing the intensity of the relationship between the values recorded by the Romanian companies and the average values of the EU, we selected seven variables (only a single most relevant variable for each pattern, namely where the highest level of the regression correlation index was registered) for which we subsequently measured the distance between the two categories of companies, a process was needed by which the calculated values of the distances for each selected variable could be aggregated. In this way, we obtained an overview picture, and different distances between variables could be reflected in a single synthetic and more representative value. In this regard, to measure the room for sustainable growth from the selected variables, the geometric mean was used. This can be found in the data envelopment analysis (DEA) for evaluating the World Knowledge Competitiveness Index (WKCI) through a quantitative analytical technique [77].

The variables taken into account are reflected differently in the geometric mean: those recording lower values in Romania compared to the EU, for which there is room for improvement (lower variables), are filled in as the numerator. The variables that, for Romania, record values above

the EU average (higher variables), which serve to compensate the deficit of lower variables, are filled in as the denominator. The calculation formula is shown below in Equation (3):

$$D_{GM}(Vj) = \sqrt[m]{\frac{\prod_{a=1}^{l} D(Vj)_{low_a}}{\prod_{b=1}^{p} D(Vj)_{high_b}}} \tag{3}$$

where

- $D_{GM}$ is the distance calculated based on the geometric mean $GM$ in a cumulative manner, of all the individual distances of the selected variables $D(Vj)$;
- $D(Vj)_{low_a}$ is the value of the calculated distances for lower values of the selected variables recorded in Romania compared to the EU average;
- $D(Vj)_{high_b}$ is the value of the calculated distances for higher values of the selected variables recorded in Romania compared to the EU average;
- $a$ is the calculated distance for lower values of the selected variables recorded, where $a = \overline{1,l}$, and $l \leq j$;
- $b$—is the calculated distance for higher values of the selected variables recorded, where $b = \overline{1,p}$, and $p \leq j$.

The higher the value of the cumulatively calculated distance is, the greater the room for sustainable growth will be for Romanian accommodation businesses.

## 4. Results

To assess the room for sustainable growth for Romanian accommodation businesses, we have taken into consideration only those variables for which there is a reasonable level of correlation with the evolution of the EU average. The results for the relationship measurement are presented below in Table 6.

**Table 6.** Correlation testing to select the most relevant variables proposed to measure the distance for accommodation businesses from Romania and the EU average.

|  | Variables Symbol | Relationship Measuring (Correlation Coefficient) | Representativeness for Variability (R-Squared Adjusted) | *p*-Value |
|---|---|---|---|---|
| 1 | TurnPersEmpl | 0.8913 | 78.16 | 0.0010 |
| 2 | NoUnpPersEmpl | 0.0203 | −6.21 | 0.7910 |
| 3 | NoEmpl | 0.8921 | 78.31 | 0.0113 |
| 4 | AvPersCosts | 0.9236 | 84.39 | 0.0001 |
| 5 | GrowRateEmpl | 0.1822 | −2.73 | 0.4693 |
| 6 | TotPurchGS | 0.9040 | 80.59 | 0.0009 |
| 7 | GrossOperSurpl | 0.4837 | 18.61 | 0.0420 |
| 8 | GrossOpRate | 0.0072 | −6.24 | 0.9774 |
| 9 | ShareGrossOpSurplValAdd | 0.0467 | −6.02 | 0.8539 |
| 10 | ProdVal | 0.9431 | 82.25 | 0.0000 |
| 11 | AppLabProduct | 0.8045 | 62.53 | 0.0001 |
| 12 | SharePersCostProd | 0.2607 | 0.97 | 0.2960 |
| 13 | SharePersCostPurch | −0.0146 | −6.23 | 0.9541 |
| 14 | InvestPersEmpl | 0.8163 | 64.55 | 0.0000 |
| 15 | InvestRate | 0.6806 | 42.96 | 0.0019 |
| 16 | ValAddFactCost | 0.8331 | 67.50 | 0.3578 |
| 17 | GrossValAddEmpl | 0.7821 | 58.75 | 0.0001 |
| 18 | ValAddProdVal | 0.2303 | −0.61 | 0.3578 |

Source: Processing performed by the authors.

After performing correlation testing, the most relevant variables selected in order to assess sustainable growth are shown below in Table 7.

**Table 7.** Quantitative variables selected for each pattern to assess sustainable growth after regression correlation.

| Patterns for Assessing/Measuring Business Growth | Quantitative Variables Proposed and Selected as Being the Most Relevant for Each Pattern |
|---|---|
| Sales growth | Turnover per person employed |
| Employee growth | Average personnel costs |
| Indebtedness level (equity, liabilities) | Total purchases of goods and services |
| Financial performance (profits, profitability) | Gross operating surplus |
| Asset value, productivity, fixed costs, liquidity | Production value |
| Investments | Investment per person employed |
| Economic value added | Value added at factor cost |

Source: Processing performed by the authors.

It is significant that for all the variables that have registered meaningful values of the correlation coefficient, these values have been positive. The evaluation of distance has been performed only for those variables that have reached a reasonable level of representativeness for each pattern.

Distance measurement will be conducted for the 7 selected variables out of the 18 analyzed: turnover per person employed, average personnel costs, total purchases of goods and services, gross operating surplus, production value, investment per person employed and added value at factor cost. The results obtained by assessing the distances between Romanian accommodation businesses and the EU average level are shown below in Table 8:

**Table 8.** The distance measurement between the variables of Romanian and EU accommodation businesses.

| Selected Variables Symbol ($V_j$) | Distance Measured $D(V_j)$ (%) | Romanian Level Compared with EU for Accommodation Businesses |
|---|---|---|
| 1 | TurnPersEmpl | 65.07 | Lower |
| 2 | AvPersCosts | 79.12 | Higher |
| 3 | TotPurchGS | 82.15 | Lower |
| 4 | GrossOperSurpl | 83.39 | Lower |
| 5 | ProdVal | 86.22 | Lower |
| 6 | InvestPersEmpl | 37.02 | Lower |
| 7 | ValAddFactCost | 87.69 | Lower |

Source: Processing performed by the authors.

Aggregating, by using the geometric mean, the negative distances calculated for variables which record values below the EU average in Romania (lower level variables) and the positive distances related to higher values in Romania compared to the EU average (higher level variables) leads to a figure of 20.6% for the sustainable growth perspective. Therefore, the distance between the Romanian and the EU average in terms of accommodation businesses is 20.6%.

## 5. Discussions

One can notice that the sustainable growth of Romanian accommodation businesses is deficient in terms of other methods of estimating it rather than in measuring its turnover [14]. These variables refer to production values, added value at factor, costs and total purchases of goods and services, and gross operating surplus (the relative distance to the EU average being over 80%). Regarding these parameters, Romanian tourism is at 20.6% of the average level achieved in the last 18 years in the EU.

In the light of these findings, some of the variables included in the study were based on the size of profit, which has a vital role in a company's growth [15,45–47].

The best values for accommodation businesses in Romania have been registered for average personnel costs, standing at 20.88% of the EU average (the relative distance to the EU average is 79.12%). This is the strength that generates sustainable growth for Romanian accommodation businesses. These variables related to employees are highlighted by the literature when assessing sustainable growth [29].

Also, investments per person employed in the Romanian tourism industry are below those registered in the EU in terms of investment rates. Investments in accommodation businesses in Romania are lower by 37.02% in comparison to the average recorded in the EU. In this case, the distance from the European Union average is not as great as for the other variables for which deficits have been recorded. From this point of view, it can be stated that investments provide more sustainable recovery for growth [68]. These parameters, when individually assessed, provide a partial picture, from a certain standpoint, of Romanian tourism in relation to EU tourism. To have a consistent, overall view for achieving a figure for a sustainable growth perspective, the aggregation of these values is required.

Given that the number of Romanian accommodation businesses increased by 7.31 times, reaching 47.03% of the EU average, while the average number of accommodation businesses from the EU remained relatively constant for the 1999–2016 period, an increase of 20.6% was recorded in terms of the sustainable growth perspective, by reference to the current EU average for this area of tourism; i.e., accommodation. In other words, based on the variables taken into consideration (those which present a reasonable level of correlation between the values of Romania and the EU average), after applying the distance method and the aggregation method [76,77], the accommodation sector in Romania still has a sustainable growth potential of 20.6%, given that now it is at a level of 79.4%, compared to the EU. Romanian accommodation businesses are smaller than the European competitors. If we take into account the fact that small firms grow faster than big firms [51], then the possibility of recovering this deficit is realistic.

The strength that has fueled the current sustainable growth potential of Romanian accommodation businesses is the reduced level of the average personnel costs of staff.

Management decisions to be taken into consideration in order to support sustainable growth in the Romanian accommodation business area refer to the variables where a deficit was recorded. Turnover per person employed is very low compared to the average of the European Union. To improve this ratio, it is necessary to increase the efficiency of staff, a measure that should be considered for the sustainable growth of accommodation businesses in Romanian tourism. In particular, it is necessary to increase each employee's contribution to turnover. This rise in efficiency is made possible by increasing the average level of investment made for each employee. On the other hand, average personnel costs are below the EU level. From this perspective, it is clear that an increase in staff efficiency is supported by the social conditions of workers, which require an additional measure: an increase in staff remuneration. Under these conditions, prerequisites will be created to achieve higher added value.

Thus, sustainable growth in Romanian businesses within the accommodation area can be promoted by managerial decisions taken at the same time as raises in salaries and investments. The impact of this will be on value-added growth and, ultimately, on the turnover achieved.

## 6. Conclusions

The literature relates to two types of variables pertaining to business growth: those that condition growth and those that assess/measure achieved growth [54]. These studies verify the existence of a correlation between the two types of variables to try to determine which one conditions or supports growth in companies, measured under different aspects.

Companies' growth can be measured by various patterns, such as sales, employees, profits, investments and economic value added. In order to assess the sustainable room for growth, this study takes into account these quantitative variables, reflected by the annual financial reports of tourism

companies that operate in the Romanian and EU accommodation area: hotels, camping sites and other forms of provision of short-stay accommodation. To assess sustainable growth, the most adequate option has been used; namely a comparative analysis of the evolution in time for a period of 18 years, in the 1999–2016 interval.

The analysis has been conducted between the Romanian and EU average per country of accommodation businesses. The study includes an average annual number of 3447 Romanian companies and an average of 9977 EU companies per country from the accommodation area.

A total of 18 quantitative variables have been selected for both groups of companies through financial reports, and for seven of them, a significant and representative level of correlation has been determined through regression analysis.

This study covers a gap in the literature, as it suggests a method of comparing the growth recorded for two categories of companies in order to assess the sustainable growth capacity of the less effective category of companies until they reach the growth level of the category of effective companies. To assess the difference between the two categories of companies, the distance method has been used [76]. In the end, in order to obtain a uniform image, the method of the geometric mean has been used for aggregation [77].

This research enhances the literature through the provision of a means of assessing sustainable growth by measuring the difference between two categories of companies. The patterns used in this respect are considered appropriate for growth measurement by the literature. The study has measured the distance between each of these patterns for both categories of accommodation businesses. In the end, the study has presented an aggregation method for putting together all the distances previously assessed between each pattern. Thus, we have reached a final conclusion related to the difference in sustainable growth. The results show the difference in growth for both categories of companies. This difference is to be considered for the second type of company ranked as having its own room for sustainable growth.

The outputs show that Romanian accommodation businesses have a sustainable growth capacity of 20.6% compared to the average growth recorded in the EU by the same categories of companies. These conclusions are reached considering the fact that, during the 18 years analyzed, the number of Romanian tourism companies increased by more than 7 times and the average number of employees for each tourism company decreased by 76.32%, faster than the EU average, which was 33.33%. One can extract complementary aspects from the study about the performance of Romanian companies related to low staff costs. This study also outlines the aspects that need to be improved in terms of production, purchases, employment, the number of employees, turnover, value added and productivity.

For Romanian accommodation companies to achieve growth, as shown in the current research, measures should be taken in order to improve ratios such as the turnover per person employed, total purchases of goods and services, and value added at factor cost. It is quite complicated for corporate governance to take, in a short period of time, such a large number of decisions to cover all these deficient issues. Further analyses of the current research are expected to cover a multivariate analysis of variance so as to emphasize the minimum combination of decisions that should be taken to enhance these ratios in order to achieve the maximum sustained growth effect over time.

**Author Contributions:** Conceptualization, S.-R.B. and A.-G.C.; Methodology, S.-R.B. and A.-G.C.; Software, S.-R.B.; Validation, S.-R.B. and A.-G.C.; Formal Analysis, S.-R.B.; Investigation, S.-R.B.; Resources, S.-R.B. and A.-G.C.; Data Curation, S.-R.B.; Writing—Original Draft Preparation, S.-R.B.; Writing—Review & Editing, A.-G.C.; Visualization, S.-R.B. and A.-G.C.; Supervision, S.-R.B. and A.-G.C.; Project Administration, S.-R.B.; Funding Acquisition, S.-R.B.

**Funding:** This research was funded by "Babeş-Bolyai" University Romania, grant number 31781/23.03.2016.

**Acknowledgments:** This work was supported by "Babeş-Bolyai" University Romania, through the Funding contract for the implementation of grants for young researchers; project number 31781/23.03.2016 "Corporate governance particularities and their impact on financial statements".

**Conflicts of Interest:** The authors declare no conflict of interest.

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
