# Peer review of "Assessing the Sustainable Room for Growth for a Particular Romanian Tourism Area of Business: The Case of Accommodation Businesses"

_sustainability, doi:10.3390/su11010243_

Round 1
Reviewer 1 Report
This is a good paper for Sustainability, although it needs certain improvement before acceptance. My recommendations are given below.
- title: the expression "the room for the growth" should be replaced with more exact term(s), why accommodation (hospitality industry) is mixed with tourism industry – please, normalize;
- the issue(s) of sustainability, sustainable development, sustainable socio-economic growth, etc. should be reflected well in the title, abstract, and the text, and I also encourage the authors to cite the basic literature on these issues, e.g.:
Karppi, I., Kultalahti, O. & Kultalahti, J. (2012). On socio-economic sustainability and robustness. European Spatial Research and Policy, 19, 5-7.
Iribarren, D., Martin-Gamboa, M., O'Mahony, T. & Dufour, J. (2016). Screening of socio-economic indicators for sustainability assessment: a combined life cycle assessment and data envelopment analysis approach. International Journal of Life Cycle Assessment, 21, 202-214.
- abstract: please, make it less declarative and more informative (e.g., indicate your recommendations);
- if table 1 is the author's own development, why not to place it in Results and provide the relevant methodological explanations?
- if the literature review is so extensive, why not to separate it from Introduction as a particular section?
- why does not the available literature review mention the tourism and hospitality industries?
- Results should be separated from Discussion, and Discussion should contain enough citations to the available literature, I also encourage the authors to try to compare their really interesting and important findings with those for some other countries;
- managerial implications should be explained with more details;
- the style of citations should be updated according to the standard MDPI's requirements;
- the language is generally ok, but the authors should re-read their paper twice to polish phrases.
Author Response
Response to Reviewer 1:
Thank you for taking the time to review this paper and for your valuable recommendations. We have addressed all your comments, as described below, while making all the recommended changes, in order to increase the potential of this paper for publication.
Reviewer’s comment 1: The expression "the room for the growth" should be replaced with more exact term(s), why accommodation (hospitality industry) is mixed with tourism industry – please, normalize
Authors’ response: According to your suggestions, we have revised our paper and made a clearer distinction between the items related to accommodation and tourism industry. We have rephrased the title into a more explicit one, so as to emphasize that the article refers to a specific area of Romanian tourism, namely to businesses from the accommodation area.
Reviewer’s comment 2: The issue(s) of sustainability, sustainable development, sustainable socio-economic growth, etc. should be reflected well in the title, abstract, and the text, and I also encourage the authors to cite the basic literature on these issues, e.g.
Authors’ response: We appreciate and agree with your recommendations related to items of sustainability. To improve the clarity of the paper, the term sustainability is now reflected in the title, abstract, and content of the article, along with additional explanations that justify the fact that the value of the growth outlook measured by this study is sustainable: it is based on a sufficiently long timeframe, combines a pattern mix which is dedicated to the literature assessing the growth capacity of a business, for each pattern we have analyzed those variables accepted by literature and selected only the ones which are relevant according to regression analysis outputs, and at the end we have performed an aggregation of the values of these variables in order to obtain a single value representing the prospects for sustainable growth.
We appreciate and have considered your useful suggestions in term of dedicated literature concerning sustainability.
Reviewer’s comment 3: Abstract: please, make it less declarative and more informative (e.g., indicate your recommendations)
Authors’ response: We agree with your suggestions in order to improve the abstract content. To elaborate, a more informative abstract has been created, with suggestions regarding the decisions that the corporate governance of Romanian companies in the tourism sector, more precisely accommodation area businesses, should consider in order to diminish their sustainable growth deficit compared to the average level of the European Union. For Romanian businesses, sustainable growth is promoted by the low level of staff costs and, to a lesser extent, by the investments made per employed person. For the corporate governance of these companies to recover growth space in terms of sustainability, they should take managerial decisions to increase sales, profitability, production value and value added.
Reviewer’s comment 4: If table 1 is the author's own development, why not to place it in “Results” and provide the relevant methodological explanations?
Authors’ response: Table 1 is a synthesis made by authors based on data provided by Eurostat, which is the starting point of the research. Following the valuable suggestions that we have received, we have reformulated the explanations of this data to provide more clarity related to the source. According to these rephrased explanations, it is now clear that the data source is Eurostat, a Directorate-General of the European Commission.
Reviewer’s comment 5: If the literature review is so extensive, why not to separate it from Introduction as a particular section?
Authors’ response: Sustainability can be approached through a wide range of variables. Some of them support sustainability, while others measure it. To theoretically substantiate the multitude of variables that measure the sustainability of the study, one has to address a broad bibliography in the field. In the case of a comprehensive and complete approach of the specialized literature being required, it may be justified to allocate a distinct subchapter in the structure of the research. Under these circumstances, we thank the reviewer for the valuable recommendations made in this respect, which have been taken into account. As a result of these completions, the research is better structured into five parts: Introduction, Literature Review, Materials and Methods, Results and Discussions and, finally, the fifth part, Conclusions.
Reviewer’s comment 6: Why does not the available literature review mention the tourism and hospitality industries?
Authors’ response: Sustainable growth is frequently addressed in macroeconomic literature. There is a much lower number of studies that address this issue at microeconomic level. In these studies, sustainable growth is frequently approached by businesses that are active in different areas: services, manufacturing, non-manufacturing, online or listed businesses on the stock market. There are few studies specifically addressing the tourism sector from a sustainable perspective and very few of them assess sustainable growth based on financial accounting ratios. Considering this, to complete the quantification of sustainable growth for the tourism sector, and specially for accommodation businesses, it was necessary to refer to this research by using this kind of financial accounting indicators, even though in literature they focus on other areas of activity too.
We appreciate and have considered your useful suggestions in terms of dedicated literature concerning sustainability in tourism. Following these constructive suggestions, supplementary literature has been added, specifically referring to sustainability in tourism: (Modica et al., 2018; Tudorache et al., 2017; Brătucu et al., 2017, Iunius et al., 2015).
Reviewer’s comment 7: Results should be separated from Discussion, and Discussion should contain enough citations to the available literature, I also encourage the authors to try to compare their really interesting and important findings with those for some other countries
Authors’ response: We appreciate and agree with your suggestions regarding the creation of two separate sections: one for research results and another for discussions. Moreover, the discussions of results have been extended in order to improve the clarity of the paper, and bring more economic interpretations for our empirical results. The suggestions on bibliographic references to similar studies are welcome. They have been considered in the Discussions section.
Reviewer’s comment 8: Managerial implications should be explained with more details
Authors’ response: Thank you for your suggestion. We have inserted other paragraphs in the Discussions section, where we have introduced additional and more detailed suggestions related to managerial decisions that could be taken into account by managers in the case of Romanian accommodation businesses for achieving sustainable growth.
Reviewer’s comment 9: The style of citations should be updated according to the standard MDPI’s requirements
Authors’ response: According to the reviewer’s comment, the citations included in the content of the research have been updated in accordance with article editing instructions: the MDPI’s requirements.
Reviewer’s comment 10: The language is generally ok, but the authors should re-read their paper twice to polish phrases.
Authors’ response: Thank you for pointing this out. According to your suggestions, the article has been reviewed once again from the perspective of English language/style and grammar by a native English-speaking colleague from the Department of Foreign Languages of our University.
We sincerely appreciate your insightful and constructive comments and suggestions. We believe that these have greatly enhanced the quality of the paper.
Once again, thank you for taking the time to review this paper.
Sincerely,
The authors
Reviewer 2 Report
OVERALL COMMENT
This article presents an attempt to analyze the growth capacity of the tourism sector in Romania. To this end, it analyses different financial variables of the country's tourism companies and compares them with those of the European Union, obtaining a series of conclusions from the comparison of these variables.
The intention of the study seems adequate and correct and can certainly be of great interest. However, in my opinion, the use of a linear regression and the distance method seems somewhat limited and simplistic.
I believe that perhaps with a somewhat more exhaustive exploitation of the analysis and a better presentation of the results obtained, a better and more interesting article can be generated.
SPECIFIC COMMENTS
1.- Perhaps the editor can comment better on this aspect, but I think the references are not in the correct format for publication.
2.- The format used in the different tables of the document does not help in understanding the information. I suggest reformatting the information in the tables in a way that is somewhat more understandable and maintaining a coherent format (font size, spacing, titles...) in all of them.
3.- Table 1: Could it be possible to add the information about the number of employees and the number of accommodation units to this table? I think it would help, as well as a figure representing the evolution of these variables.
4.- Introduction. I suggest creating a new section “2. Literature review” starting on line 71, and adding a short paragraph stating the objective of the paper before line 68.
5.- Line 55: references are needed.
6.- Table 6: Does “confidence level” refers to “p-value”? Could other statistics be obtained? F-statistics, log likelihood… I am not sure if the column “The variables relationship intensity” is necessary. Anyway, how is this intensity obtained?
7.- What is the model used for the regression? It must be described (equation included) somewhere in the text.
8.- Can the results obtained for Romania be compared to other similar studies within the EU? I.E. results from Table 8, are they high? Low?
Author Response
Response to Reviewer 2:
Thank you for taking time to review this paper and for your valuable recommendations. We have addressed all your comments, as described below, while making all the recommended changes, in order to increase the potential of this paper for publication.
Reviewer’s comment 1: Perhaps the editor can comment better on this aspect, but I think the references are not in the correct format for publication
Authors’ response: Following the reviewer’s comment, the citations included in the content of the research have been updated in accordance with article editing instructions: the MDPI’s requirements.
Reviewer’s comment 2: The format used in the different tables of the document does not help in understanding the information. I suggest reformatting the information in the tables in a way that is somewhat more understandable and maintaining a coherent format (font size, spacing, titles...) in all of them.
Authors’ response: We have made reformulations of the table headings and, consequently, simplifications have been made to table contents following these rewritings. After considering the reviewer's suggestions, the information that is intended to be transmitted has become more intelligible.
Reviewer’s comment 3: Table 1: Could it be possible to add the information about the number of employees and the number of accommodation units to this table? I think it would help, as well as a figure representing the evolution of these variables.
Authors’ response: The number of companies in the Romanian business sector was included in Table 1. Eurostat can provide data on the total number of employees and companies in the field of accommodation for the EU and this data could be tabled in accordance with the nature of the study. We appreciate the interest in this additional data, but the relevance would be low as the study is based on a comparison between the Romanian and EU average (in this research, the EU average is considered as that of a stand-alone country, used as the basis for comparison with the level recorded in Romania). According to the reviewer's constructive suggestions, two figures have been drawn up based on Table 1, so as to better explain the circumstances underlying the present research.
Reviewer’s comment 4: Introduction. I suggest creating a new section “2. Literature review” starting on line 71, and adding a short paragraph stating the objective of the paper before line 68
Authors’ response: Sustainability can be approached through a wide range of variables. Some of them support sustainability, while others measure it. To theoretically substantiate the multitude of variables that measure the sustainability of the study, one has to address a broad bibliography in the field. In the case of a comprehensive and complete approach of the specialized literature being required, it may be justified to allocate a distinct subchapter in the structure of the research. Under these circumstances, we thank the reviewer for the valuable recommendations made in this respect, which have been taken into account. As a result of these completions, the research is better structured into five parts: Introduction, Literature Review, Materials and Methods, Results and Discussions and, finally, the fifth part, Conclusions.
Reviewer’s comment 5: Line 55: references are needed
Authors’ response: The above-mentioned paragraph refers to the fact that “Relevant literature includes studies that test by various methods the correlation between a certain category of parameters which determine growth and another category of parameters that measure growth itself, as they are mentioned above”. We appreciate and agree with your comment in this regard. According to your suggestions, we have revised this paragraph, and supporting references as literature review have been provided.
Reviewer’s comment 6: Table 6: Does “confidence level” refers to “p-value”? Could other statistics be obtained? F-statistics, log likelihood… I am not sure if the column “The variables relationship intensity” is necessary. Anyway, how is this intensity obtained?
Authors’ response: According to your suggestions, we have revised the structure and content of Table 6, and have provided a clearer explanation for the meaning of confidence level. This is explained by the level of statistical significance of the relationship between the dependent and independent variable, based on the level of P-value in the ANOVA table. The column related to variable relationship intensity has been removed. You are perfectly right: the explicit presentation of variables relationship intensity is not necessarily required as long as the level of this intensity is implicitly derived from the first column, where the correlation coefficient level was established. We are on the point of continuing our research and intend to apply additional statistical indicators in the next study that we will develop on this subject in the near future.
Reviewer’s comment 7: What is the model used for the regression? It must be described (equation included) somewhere in the text.
Authors’ response: To improve the clarity of the paper, it is indeed necessary to describe the regression model used. In the case of the present study, it is the linear regression model. According to the valuable suggestions provided, the model equation has been included in the Materials and Methods section of the article.
Reviewer’s comment 8: Can the results obtained for Romania be compared to other similar studies within the EU? I.E. results from Table 8, are they high? Low?
Authors’ response: We appreciate and agree with your suggestions on comparing the results achieved for Romania with other similar studies conducted. This will provide more consistency to the research. Therefore, the discussions of results have been extended in order to improve the clarity of the paper, and bring more economic interpretations for our empirical results. Suggestions on bibliographical references to outputs of similar studies are welcome. They have been considered in the Discussions section.
We sincerely appreciate your insightful and constructive comments and suggestions. We believe that these have greatly enhanced the quality of the paper.
Once again, thank you for taking the time to review this paper.
Sincerely,
The authors
Reviewer 3 Report
The subject of the article is interesting.
The purpose of the work should be clearly defined.
In abstract, the aim was defined as: "… to evaluate whether there is still room for sustainable growth", and at the paper there is nothing about sustainable development.
Literature review is not fully related to the research problem. There is nothing about tourist enterprises and sustainable development.
I suggest re-editing the title of the study, for example, "The possibility of growth in accommodation businesses within the Romanian tourism industry" or other.
Data from table 1 should be presented in the form of figure (this will improve their readability).
Wers 55 "previously published research" and 56 "relevant literature". Please write what literature is meant. Who? Where?
Verses 55-59 – very general, it does not add to the text.
Verses 76-77 "OECD countries over the past 140 years". Which research, please explain.
Please standardize the literature footnotes. The work requires editorial elaboration.
The conclusions do not refer to the aim of the paper.
It is necessary to pay more attention to the aspect of sustainable development at the paper (literature review, discussion, conclusions).
Author Response
Response to Reviewer 3:
Thank you for taking time to review this paper and for your valuable recommendations. We have addressed all your comments, as described below, while making all the recommended changes, in order to increase the potential of this paper for publication.
Reviewer’s comment 1: The purpose of the work should be clearly defined
Authors’ response: We have considered you valuable suggestion on improving the research purpose. According to your comment, we have revised our research purpose and provided a more explicit description within the Introduction section of the paper.
Reviewer’s comment 2: In abstract, the aim was defined as: "… to evaluate whether there is still room for sustainable growth", and at the paper there is nothing about sustainable development.
Reviewer’s comment 11: It is necessary to pay more attention to the aspect of sustainable development at the paper (literature review, discussion, conclusions).
Authors’ response: We appreciate and agree with your recommendations related to items of sustainability. Thus, to improve the clarity of the paper, the term sustainability has been reflected in the title, abstract, and content of the article, along with additional explanations that justify the fact that the value of the growth outlook measured by this study is sustainable: it is based on a sufficiently long timeframe, combines a pattern mix which is dedicated to the literature on assessing the growth capacity of a business, for each pattern we have analyzed those variables accepted by literature and selected only those which are relevant according to regression analysis outputs, at the end we have performed an aggregation of the values of these variables in order to have a single value representing the prospects for sustainable growth.
Reviewer’s comment 3: Literature review is not fully related to the research problem. There is nothing about tourist enterprises and sustainable development.
Authors’ response: Sustainable growth is frequently addressed in macroeconomic literature. There is a much lower number of studies that address this issue at microeconomic level. In these studies, sustainable growth is frequently approached by businesses which are active in different fields of activity: services, manufacturing, non-manufacturing, online or listed businesses on the stock market. There are few studies specifically addressing the tourism sector from a sustainable perspective and very few of them assess sustainable growth based on financial accounting ratios. Considering this, to complete the quantification of sustainable growth for the tourism sector, and specially for accommodation businesses, it has been necessary to refer to research using this kind of financial accounting indicators, even though in literature they focus on other areas of activity too.
We appreciate and have considered your useful suggestions in terms of dedicated literature concerning sustainability in tourism. Following these constructive suggestions, supplementary literature has been added, specifically referring to sustainability in tourism: (Modica et al., 2018; Tudorache et al., 2017; Brătucu et al., 2017, Iunius et al., 2015).
Reviewer’s comment 4: I suggest re-editing the title of the study, for example, "The possibility of growth in accommodation businesses within the Romanian tourism industry" or other.
Author’s response: According to your suggestions, we have revised our paper and made a clearer distinction between items related to accommodation and tourism industry. We have rephrased the title into a more explicit one, so as to emphasize that the article refers to a specific area of Romanian tourism, namely to businesses from the accommodation area.
Reviewer’s comment 5: Data from table 1 should be presented in the form of figure (this will improve their readability).
Author’s response: According to the reviewer's constructive suggestions, two figures have been drawn up based on Table 1, so as to better explain the circumstances underlying the research.
Reviewer’s comment 6: Wers 55 "previously published research" and 56 "relevant literature". Please write what literature is meant. Who? Where?
Reviewer’s comment 7: Verses 55-59 – very general, it does not add to the text.
Author’s response: The above-mentioned paragraph refers to the fact that “Relevant literature includes studies that test by various methods the correlation between a certain category of parameters which determine growth and another category of parameters that measure growth itself, as they are mentioned above”. We appreciate and agree with your comment in this regard. According to your suggestions, we have revised this paragraph and removed it. The content of these remarks has been considered in the Results and Discussions sections of the paper.
Reviewer’s comment 8: Verses 76-77 "OECD countries over the past 140 years". Which research, please explain.
Author’s response: We have revised this paragraph. The information provided does not fit into the context, which is why we have removed it. Thank you for pointing this out.
Reviewer’s comment 9: Please standardize the literature footnotes. The work requires editorial elaboration.
Author’s response: According to the reviewer’s comment, the citations included in the content of the research and the literature footnotes have been updated in accordance with article editing instructions: the MDPI’s requirements.
Reviewer’s comment 10: The conclusions do not refer to the aim of the paper.
Author’s response: We appreciate and agree with your suggestions. We have inserted other paragraphs in the Conclusions section, where we have rephrased and included additional information for more relevance of the conclusions.
We sincerely appreciate your insightful and constructive comments and suggestions. We believe that these have greatly enhanced the quality of the paper.
Once again, thank you for taking the time to review this paper.
Sincerely,
The authors
Reviewer 4 Report
I think that this paper is not related with sustainability and I guess that this paper will not attract a broad interest readership of Sustainability.
Author Response
Response to Reviewer 4:
Thank you for taking the time to review this paper and for your valuable recommendations. We have addressed all your comments, as described below, while making all the recommended changes, in order to increase the potential of this paper for publication.
Reviewer’s comment 1: I think that this paper is not related with sustainability and I guess that this paper will not attract a broad interest readership of Sustainability.
Authors’ response:
To elaborate, a more informative abstract has been created, with suggestions regarding the decisions that the corporate governance of Romanian companies in the tourism sector, more precisely accommodation area businesses, should consider in order to diminish their sustainable growth deficit compared to the average level of the European Union. For Romanian businesses, sustainable growth is promoted by the low level of staff costs and, to a lesser extent, by the investments made per employed person. For the corporate governance of these companies to recover growth space in terms of sustainability, they should take managerial decisions to increase sales, profitability, production value and value added.
We have considered your suggestion on improving the research purpose. According to your comment, we have revised our research purpose and provided a more explicit description within the Introduction section of the paper. Two figures have been drawn up based on Table 1, so as to better explain the circumstances underlying the present research.
To improve the clarity of the paper, the term sustainability is now reflected in the title, abstract, and content of the article, along with additional explanations that justify the fact that the value of the growth outlook measured by this study is sustainable: it is based on a sufficiently long timeframe, combines a pattern mix which is dedicated to the literature assessing the growth capacity of a business, for each pattern we have analyzed those variables accepted by literature and selected only the ones which are relevant according to regression analysis outputs, and at the end we have performed an aggregation of the values of these variables in order to obtain a single value representing the prospects for sustainable growth.
We have inserted other paragraphs in the separate Discussions section, where we have introduced additional and more detailed suggestions related to managerial decisions that could be taken into account by managers in the case of Romanian accommodation businesses, in order to achieve sustainable growth.
We have compared the results for Romania with other similar studies conducted. This will provide more consistency to the research. Moreover, the discussions of the results have been extended so as to improve the clarity of the paper, and bring more economic interpretations for our empirical results. Suggestions on bibliographic references to outputs of similar studies are welcome. They have been considered within the Discussions section.
We have inserted other paragraphs in the Conclusions section, where we have rephrased and included additional information to give more relevance to the conclusions.
We believe that these constructive comments have greatly enhanced the quality of the paper.
Once again, thank you for taking the time to review this paper.
Sincerely,
The authors
Round 2
Reviewer 1 Report
I am fully satisfied with how the authors corrected the text and responded to the reviewers.
Author Response
Response to Reviewer 1 Round 2:
We appreciate your thorough evaluation and have found your recommendations to be very helpful. We have carefully addressed the point you have mentioned, in order to enhance the quality of the article and its compatibility with the Journal.
Reviewer recommendation: English language and style are fine/minor spell check required
Authors’ answer: We have once again reviewed the text from the linguistic standpoint and have rephrased several sentences, so as to clarify their meaning and context.
Once again, thank you for your time and consideration.
Sincerely,
The authors
Reviewer 2 Report
Dear authors.
I am grateful for the comments and modifications made by the authors as a result of the review carried out.
The manuscript has been substantially improved, however there are still aspects that in my opinion need to be addressed and improved to make the paper finally publishable.
- When referring to a figure, please always state its number: i:e. in lines 45 or 54. “Above”, “previous”, “next” should not be used.
- I strongly recommend changing the values in column “Confidence level (p-value)” on Table 6 to those obtained from the studied developed. I do not understand why to limit this information to “higher of lower than 95%” when the specific value can be given. In some cases, variables with a 94% p-value can be as interesting to the analysis as those with a 95%.
- I still believe that the analysis of the data can be more complete if a statistical analysis is carried out that goes beyond a simple univariate linear regression. Has the possibility of a multivariate analysis been studied?
- Discussion and Conclusions are now much interesting than in the first version. But I suggest enhancing them with further analysis. If as the authors have said, they are working in further research and analysis of available data and relationships between variables, maybe some preliminary results can be included in this paper.
- Error with bookmark on line 375.
Author Response
Response to Reviewer 2 Round 2:
We appreciate your thorough evaluation and have found your recommendations to be very helpful. We have carefully addressed all the points you have mentioned, in order to enhance the quality of the article and its compatibility with the Journal.
Reviewer’s comment 1: When referring to a figure, please always state its number: i.e. in lines 45 or 54. “Above”, “previous”, “next” should not be used.
Author’s response: We appreciate and agree with your suggestions. To improve the clarity of the research, we have introduced an explicit reference to Figure 1 in the content of the article. For added clarity, we have replaced, according to your valuable suggestion, the word “figure” in line 54 with the more appropriate word “ratio”.
Reviewer’s comment 2: I strongly recommend changing the values in column “Confidence level (p-value)” on Table 6 to those obtained from the studied developed. I do not understand why to limit this information to “higher of lower than 95%” when the specific value can be given. In some cases, variables with a 94% p-value can be as interesting to the analysis as those with a 95%.
Author’s response: Thank you for your suggestions. We have inserted replacing figures in Table 6 according to the levels achieved for P-value in the ANOVA table, in order to emphasize more accurately the statistical significance relationship between the variables analyzed.
Reviewer’s comment 3: I still believe that the analysis of the data can be more complete if a statistical analysis is carried out that goes beyond a simple univariate linear regression. Has the possibility of a multivariate analysis been studied?
Author’s response: We appreciate your suggestions related to the statistical analysis carried out. The authors are indeed considering the possibility of expanding the research results. In this respect, complementary to the methods specific to simple regression analysis, the multivariate analysis of the data is being taken into account and, in addition to multiple regression analysis, statistical processing will be tested to extract additional conclusions. Thus, the discriminant analysis method and multivariate analysis of variance will be tested. This latter method verifies the separation and testing of the significance of the effects on sustainable growth caused by the simultaneous action of several factors from those analyzed in the first part of the research.
During the period of expansion of the research, EUROSTAT is going to publish additional data for the years 2017 and 2018. Therefore, the period of time taken into account for the research will be extended to 20 years, which will bring more relevance to the research results and will enhance the study.
Reviewer’s comment 4: Discussion and Conclusions are now much interesting than in the first version. But I suggest enhancing them with further analysis. If as the authors have said, they are working in further research and analysis of available data and relationships between variables, maybe some preliminary results can be included in this paper.
Author’s response: Thank you for pointing this out. Thanks to your valuable suggestion, the research has now been enhanced. A paragraph related to this aspect has been inserted into the Conclusion section, as follows:
“For Romanian accommodation companies to achieve growth, as shown in the current research, measures should be taken in order to improve ratios such as turnover per person employed, total purchases of goods and services, and value added at factor cost. It is quite complicated for corporate governance to take, in a short period of time, such a large number of decisions to cover all these deficient issues. Further analyses of the current research are expected to cover a multivariate analysis of variance so as to emphasize the minimum combination of decisions that should be taken to enhance these ratios in order to achieve the maximum sustained growth effect over time”.
Reviewer’s comment 5: Error with bookmark on line 375.
Author’s response: According to the reviewer’s comment, the error has been corrected and the link restored.
Reviewer’s additional recommendation: English language and style are fine/minor spell check required
Authors’ answer: We have once again reviewed the text from the linguistic standpoint and have rephrased several sentences, so as to clarify their meaning and context.
The authors express their consideration for the constructive remarks addressed by the reviewer.
Sincerely,
The authors
Reviewer 3 Report
I accept the uploaded version of the article for publication.
According to me, the trend functions are unnecessarily added, in particular in Figure 1b for Romania, the linear trend incorrectly describes the data.
Author Response
Response to Reviewer 3 Round 2:
We appreciate your thorough evaluation and have found your recommendations to be very helpful. We have carefully addressed the point you have mentioned, in order to enhance the quality of the article and its compatibility with the Journal.
Reviewer’s comment 1: According to me, the trend functions are unnecessarily added, in particular in Figure 1b for Romania, the linear trend incorrectly describes the data.
Authors' response: Thank you for pointing this out. According to the reviewer’s suggestions, the inappropriate linear trend has been removed from Figure 1b. It has been replaced by the exponential trend, which is more relevant.
The authors express their consideration for the constructive remarks addressed by the reviewer.
Sincerely,
The authors
Reviewer 4 Report
My decision is to reject this manuscript. It is not enough instead growth to use sustainable growth. Authors absolutely do not know what sustainable growth is. Authors stated that “sustainable growth is promoted by the low level of staff costs”. I do not agree with this statement. The social conditions of workers are more important than staff cost.
Furthermore, authors declared that “Romanian accommodation businesses could grow sustainably when the average level recorded for accommodation businesses at European Union level is considered to be the ideal standard”. I do not agree with this statement as well, because the sustainability it is not the average level of EU variables.
Author Response
Response to Reviewer 4 Round 2:
We appreciate your thorough evaluation and have found your recommendations to be very helpful. We have carefully addressed the point you have mentioned, in order to enhance the quality of the article and its compatibility with the Journal.
Reviewer’s comment 1:
a) My decision is to reject this manuscript. It is not enough instead growth to use sustainable growth. Authors absolutely do not know what sustainable growth is.
b) Authors stated that “sustainable growth is promoted by the low level of staff costs”. I do not agree with this statement. The social conditions of workers are more important than staff cost.
Authors’ response:
a) For growth to be sustainably assessed, it should be addressed from as many points of view as possible. This is why the study presents a complex literature review of studies which refer to growth. In these studies, there is a differentiation between aspects that sustain or condition sustainable growth and aspects that assess it after it has been accomplished. The measurement/ assessment of growth is therefore made afterwards, and is addressed in literature starting from one or more of the following perspectives: sales growth and employee growth, profitability, asset value and productivity, investments and economic value added. The growth assessed by this research is considered to be sustainable because it is assessed at the same time in a single research that includes all of these indicators put together by aggregation. Additional relevance to the measured sustainable growth is brought about by also considering in the research the aggregation of the value of these seven patterns, based on public data collected from the European Union’s Statistical Office (EUROSTAT), which covers a time span of 18 years.
b) As costs tend to be lower, there are prerequisites for the sustainable growth of any business. Indeed, the authors agree with the reviewer's view that a higher level of wage costs may be encouraging for employees. That is why in the final part of the discussion section it is mentioned in the article that the decisions of management to achieve sustainable growth must consider the increase in personnel remuneration, which in turn will generate an efficient staff, with a favorable impact on the increase in added value and, implicitly, on sustainable growth. For more clarity in this respect, the authors have taken into account the reviewer's additions and have explicitly mentioned that the proposed increase in staff remuneration leads to improved social conditions, which will later support staff efficiency. In order for Romanian accommodation companies to accumulate growth, as shown in the current research, corporate governance decisions should be taken to improve ratios such as turnover per person employed, staff costs, total purchases of goods and services, and value added at factor cost. It is quite complicated for corporate governance to take, in a short period of time, such a large number of decisions to cover all these deficient issues. That is the reason why further research will consider using multivariate analysis of variance in order to test which decision fits better for improving growth more efficiently.
Reviewer’s comment 2: Furthermore, authors declared that “Romanian accommodation businesses could grow sustainably when the average level recorded for accommodation businesses at European Union level is considered to be the ideal standard”. I do not agree with this statement as well, because the sustainability it is not the average level of EU variables.
Authors’ response: The authors agree with reviewer’s opinion that sustainability cannot be considered as the average level achieved by the European Union. Moreover, even this average level achieved by the European Union may change in the future.
The study aims to establish a current average level achieved by the European Union. This is done by aggregating the seven models that assess growth, selected according to the studies in literature and the statistical elaborations made by the authors. This established level is considered as a standard. The article then measures how much Romanian accommodation businesses could sustainably grow to reach this standard from the level at which they are now.
The authors express theirs consideration for the constructive remarks addressed by the reviewer.
Sincerely,
The authors